# Potentially Toxic Planktic and Benthic Cyanobacteria in Slovenian Freshwater Bodies: Detection by Quantitative PCR

**DOI:** 10.3390/toxins13020133

**Published:** 2021-02-11

**Authors:** Maša Zupančič, Polona Kogovšek, Tadeja Šter, Špela Remec Rekar, Leonardo Cerasino, Špela Baebler, Aleksandra Krivograd Klemenčič, Tina Eleršek

**Affiliations:** 1Department of Genetic Toxicology and Cancer Biology, National Institute of Biology, 1000 Ljubljana, Slovenia; tina.elersek@nib.si; 2Jozef Stefan International Postgraduate School, 1000 Ljubljana, Slovenia; 3Department of Biotechnology and Systems Biology, National Institute of Biology, 1000 Ljubljana, Slovenia; polona.kogovsek@nib.si (P.K.); spela.baebler@nib.si (Š.B.); 4Slovenian Environment Agency, 1000 Ljubljana, Slovenia; tadeja.ster@gov.si (T.Š.); spela.remec-rekar@gov.si (Š.R.R.); aleksandra.krivograd-klemencic@gov.si (A.K.K.); 5Department of Sustainable Agro-Ecosystems and Bioresources, Research and Innovation Centre, Fondazione Edmund Mach, 38010 San Michele all’Adige, Italy; leonardo.cerasino@fmach.it

**Keywords:** cyanotoxin detection, harmful cyanobacterial blooms, next-generation biomonitoring, real-time PCR, qPCR, LC-MS/MS, microcystin, cylindrospermopsin, saxitoxin

## Abstract

Due to increased frequency of cyanobacterial blooms and emerging evidence of cyanotoxicity in biofilm, reliable methods for early cyanotoxin threat detection are of major importance for protection of human, animal and environmental health. To complement the current methods of risk assessment, this study aimed to evaluate selected qPCR assays for detection of potentially toxic cyanobacteria in environmental samples. In the course of one year, 25 plankton and 23 biofilm samples were collected from 15 water bodies in Slovenia. Three different analyses were performed and compared to each other; qPCR targeting *mcyE*, *cyrJ* and *sxtA* genes involved in cyanotoxin production, LC-MS/MS quantifying microcystin, cylindrospermopsin and saxitoxin concentration, and microscopic analyses identifying potentially toxic cyanobacterial taxa. qPCR analyses detected potentially toxic *Microcystis* in 10 lake plankton samples, and potentially toxic *Planktothrix* cells in 12 lake plankton and one lake biofilm sample. A positive correlation was observed between numbers of *mcyE* gene copies and microcystin concentrations. Potential cylindrospermopsin- and saxitoxin-producers were detected in three and seven lake biofilm samples, respectively. The study demonstrated a potential for cyanotoxin production that was left undetected by traditional methods in both plankton and biofilm samples. Thus, the qPCR method could be useful in regular monitoring of water bodies to improve risk assessment and enable timely measures.

## 1. Introduction

Cyanobacterial blooms and a subsequent release of cyanotoxins into the environment are becoming more frequent due to eutrophication, global warming and other anthropogenic pressures. They can have negative effects on all ecosystem services as well as human and animal health and can cause economical damage by affecting tourism, recreation, industry, agriculture and drinking water supply. On the European Union level, there is no legislation prescribing regular monitoring of cyanotoxin concentration in surface waters. The most frequently used guideline is the one for drinking water from the World Health Organisation, setting the upper limit of 1 µg/L of microcystin-LR equivalents [1].

Early detection of cyanotoxin threat could help water resources managers take timely and appropriate measures. Current approaches for identification and quantification of cyanobacterial cells and cyanotoxins—microscopic count and analytical methods, such as high-performance liquid chromatography (HPLC), liquid chromatography–mass spectrometry (LC-MS), enzyme-linked immunosorbent assay (ELISA) or protein phosphatase 1A (PP1A) analyses—each have their advantages, but they can often be time-consuming, costly or technically demanding. Emerging molecular methods, such as quantitative PCR (qPCR), could enable fast, highly sensitive and cost-effective detection of potentially toxic cyanobacteria [2]. This approach is based on the extraction of community DNA from environmental samples and can therefore provide a full picture of the cyanobacterial diversity. However, for its use in regular monitoring programs, these methods have to be thoroughly tested and optimised in order to achieve comparability with current methods and thus applicability in monitoring schemes.

Analytical methods for detection of cyanotoxins can determine their concentration only at a certain point in time, whereas cyanotoxin content can vary significantly throughout the day (depending on hydrological conditions and the presence of bacterial decomposers). Besides, due to a high number of variants of cyanotoxins (e.g., over 248 variants of microcystins [3]) and a lack of standards for each of these variants, not all of them can be measured. These methods are also too expensive to monitor the concentration daily. On the other hand, detection and quantification of genes involved in cyanotoxin synthesis could enable cost-effective monitoring of the potential for cyanotoxin production on a daily basis at various locations. Moreover, all of the cyanotoxins with known genetic basis can be analysed. This can give us comprehensive information on the toxigenic potential of cyanobacterial communities in the environment.

The qPCR method has been applied in various studies, which have been summed up by Pacheco et al. [2]. Majority of the studies target genes involved in microcystin (MC) synthesis (e.g., [4], followed by cylindrospermopsin (CYN) (e.g., [5] and saxitoxin (SXT) synthesis (e.g., [6]). There have been attempts to optimise the method for its use directly in the field ([7] or to target various genes at once in multiplex reactions (e.g., [8]). However, there is still no consensus on the applicability of qPCR in regular monitoring since gene copy numbers reveal only the potential for cyanotoxin production, which is not always in correlation with actual cyanotoxin concentrations in the environment. Precisely this contrast between the methods indicates the advantage of qPCR over analytical methods focused on cyanotoxin measurement, as the former one could predict also future risk rather than assessing only the current situation.

The majority of the cyanobacterial qPCR studies are directed at microcystins, while detection of cylindrospermopsins and especially saxitoxins is still relatively rare [2]. Moreover, most of the studies focus on plankton samples, while cyanobacteria in biofilm are still underrepresented in molecular studies despite increasing evidence of their toxicity with potential acute effects on animals [9,10,11,12,13]. Furthermore, few of these studies include comparison of qPCR method and microscopy [2]. As microscopy is the preferred method of biomonitoring in many countries, it is important to investigate its effectiveness in detecting potential risk of cyanotoxin production. Additionally, all such studies are geographically limited with little or no focus on the central European region [2]. Taking into account the high genetic variability in naturally occurring cyanobacterial strains throughout the world (e.g., [14]), the assays should be tested in different regions and different water bodies to assure their wide applicability.

Therefore, the aim of this study was to expand the evaluation of qPCR assays for detection of cyanotoxin threat to understudied benthic cyanobacteria in biofilm samples, with the emphasis on comparison of results with microscopy as well as LC-MS/MS. We focused on water bodies in different regions of Slovenia (central Europe) and on three groups of cyanotoxins: microcystins, cylindrospermopsins and saxitoxins. We employed five previously published qPCR assays for detection of the microcystin- [15,16,17], cylindrospermopsin- [18] and saxitoxin-producing cyanobacteria [6]. Although the cyanotoxin potential is not necessarily linked to cyanotoxin concentration, we evaluated the correlation between the number of gene copies, microscopically determined cell number of potentially toxic species and cyanotoxin concentration. This is the first study in Slovenia aiming to detect cyanobacterial toxic potential with qPCR, and one of the few studies employing this method in environmental biofilm samples. Only thorough understanding of strengths and weaknesses of the qPCR method can enable its implementation into existing environmental monitoring strategies.

## 2. Results

### 2.1. Evaluation of the qPCR Assays

For our study, we have chosen previously published assays mcyE-Ana, mcyE-Mic, mcyE-Pla, cyrJ and sxtA, targeting microcystin-producers from genera *Dolichospermum* (ex *Anabaena*), *Microcystis* and *Planktothrix*, cylindrospermopsin-producers and saxitoxin-producers, respectively ([6,15,16,17,18]; Table 1). First, we evaluated the selected qPCR assays in terms of their specificity, sensitivity and robustness.

#### 2.1.1. Selection and Specificity of the qPCR Assays

Based on a literature review (Appendix A), we selected nine published assays for specificity evaluation. In addition to the assays shown in Table 1, two assays targeting all microcystin-producers (McyE-F2b/R4 [15,19] and DQmcy [20]), and assay anaC-gen targeting anatoxin-producers [21,22] were evaluated. Assay anaC-gen was excluded due to too high specificity indicated in the original paper [21] and too long amplicon, originally designed for end-point PCR application. The other eight assays were evaluated in vitro on test environmental samples. Assays mcyE-F2b/R4 and DQmcy were excluded based on suboptimal performance with Slovenian environmental samples (dissociation curves indicating non-target amplification (multiple peaks) and certain results inconsistent with microscopic observations, which could be due to regional differences in cyanobacterial genotypes; data not shown)).

The remaining six assays were further characterised in silico and in vitro. Specificity evaluation done in the original papers demonstrated appropriate specificity of all assays, while BLAST analysis in this study revealed that four assays (namely mcyE-Ana, mcyE-Pla, cyrJ and sxtA) are specific for desired target organisms and genes. Assays mcyE-Mic and 16S-cyano showed non-target alignment; the former with *Pseudanabaena* sp. CCM-UFV065, and the latter with plant chloroplasts as well as some heterotrophic bacteria, such as *Chryseobacterium* or *Actinobacterium*, which can be found in freshwater habitats and could thus be amplified in environmental samples. In original studies, specificity of assays mcyE-Mic [16] and 16S-cyano [6] was in vitro tested with selected non-target genera, namely *Planktothrix*, *Dolichospermum* and *Nostoc*, and with several target cyanobacterial cultures, respectively. None of the assays showed any cross-reactivity, however, unspecific reaction predicted with in silico analysis was not evaluated.

In cyanobacterial cultures, specific amplification occurred only in strains producing target cyanotoxins and not in other strains (Table 2). In assay 16S-cyano, strong amplification was observed in selected plant samples, confirming cross-reactivity with plant 16S rRNA genes (Appendix A). This could lead to false positive results and overestimation of cyanobacterial abundance in environmental samples; thus, this assay was used only for evaluation of DNA extraction and control of inhibition in qPCR reactions.

In some of the environmental samples, melting temperatures of the amplified products (Tm) obtained via dissociation curve analysis indicated the presence of non-target amplicons. This was mostly observed with assays mcyE-Ana and mcyE-Mic, which showed up to 12.3 °C and up to 10.0 °C higher Tm then the reference Tm (Table 2, Appendix A), respectively. Additionally, primer dimers were detected in some samples with assays mcyE-Ana, mcyE-Mic and mcyE-Pla (Tm < 70 °C). Therefore, results for all cyanotoxin-specific assays were considered positive if their Tm values were within the expected range (±0.5 °C). For the assay 16S-cyano, however, a wider range of obtained Tm values (80.7–82.7 °C; Appendix A) were considered positive, as they were consistent between technical replicates and the Cq values were below 30. Reference Tm values from pure cultures corresponded closely to Tm of synthetic DNA fragments for all assays (± ≤0.5 °C, data not shown). Gel electrophoresis of the samples with multiple Tm peaks (18 samples for mcyE-Ana, 3 samples for mcyE-Pla and 2 samples for cyrJ) produced multiple bands of different lengths, in contrast to environmental samples with one distinctive Tm peak that produced a single band (data not shown), confirming non-specific amplification of various DNA fragments in the former group. Therefore, such samples were considered negative.

#### 2.1.2. Sensitivity of the qPCR Assays.

Dilution series of the cyanobacterial culture DNA was prepared to evaluate the sensitivity of the assays (Table 3; calibration curves in Appendix A). All six qPCR assays showed high sensitivity, ranging from 10 to 30 cells/mL for all assays, except sxtA, which showed the highest sensitivity, detecting less than 1 cell/mL (Table 3). Amplification efficiency determined from the dilution series of all assays was between 63% and 98%.

#### 2.1.3. Robustness of the qPCR Assays

Performance of the assays was evaluated also on typical environmental samples with known cyanobacterial taxa composition (as determined by microscopy). Up to 14 samples of plankton community DNA (2 from lakes, 6 from urban ponds, 1 from urban stream and 5 cyanobacterial bloom samples) were tested with specific assays. The presence of the target organisms was mostly confirmed in samples where it was expected (Appendix A), thus proving the suitability of the method for target gene detection in environmental samples sampled in our region.

For intra-assay variability, the absolute difference between Cq values from three technical replicates in positive samples was determined. Relatively high intra-assay variability was observed in samples with target gene concentration close to LOD of the assays, which is probably due to a stochastic effect, while lower variability was observed in the rest of the samples (Appendix A). For the assay mcyE-Ana, we could not assess intra-assay variability as we did not get any positive qPCR results from the environmental samples. When using cyanobacterial monocultures as a template, the variability was in general lower than with environmental samples (Appendix A). This is expected, as the presence of inhibitory compounds in environmental samples can interfere with qPCR amplification [23], which can result in higher intra-assay variability. Inter-assay variability, evaluated for assays mcyE-Pla, cyrJ and sxtA in two separate runs, differed between assays but showed reasonably good reproducibility, taking into account degradation of DNA due to freeze/thaw cycle and different inhibitory substances in environmental samples (data not shown).

Additionally, possible inhibition of qPCR reactions was checked by testing two subsequent dilutions of 10 randomly selected samples. Only one sample showed a potential for inhibition of a qPCR reaction (BL1.5, Appendix A), therefore analysis of this sample was repeated with more diluted DNA. Robustness of the assays was demonstrated by testing DNA extracted from different matrices (cyanobacterial monocultures, synthetic DNA fragments, frozen or lyophilised bloom samples, environmental samples of plankton or biofilm), of variable purity and of variable DNA concentration (data not shown), where adequate performance was observed in all cases.

### 2.2. Presence of Target Genes in Environmental Samples

For the analysis of the environmental samples, negative and positive controls were included in every qPCR run, where the former showed no amplification and the latter showed specific amplification in all cases. Additionally, negative controls were included in every DNA extraction and in every sampling, and their NanoDrop measurements showed no presence of DNA. This confirmed appropriate assay performance and absence of contamination during field sampling, DNA extraction and preparation of qPCR reaction mixtures. Successful DNA extraction and qPCR amplification were additionally confirmed by positive results of the assay 16S-cyano amplifying 16S rRNA genes in all samples with a Cq range between 14 and 24 (Appendix A).

Detailed results of qPCR, LC-MS/MS and microscopic analyses are depicted in Appendix A, raw data is available in Appendix A. Potentially toxic *Microcystis* cells (assay mcyE-Mic) were detected in 10 lake plankton samples (all samples from Lake Vogrscek, Slivnica and Pernica). Potentially toxic *Planktothrix* cells (assay mcyE-Pla) were detected in 12 lake plankton samples (all samples from Lake Bled, in low amounts also in Lake Bohinj) and in low amounts in one biofilm sample (Lake Bled). Potentially toxic *Dolichospermum* species (assay mcyE-Ana) were not detected in any plankton nor biofilm sample. Potential cylindrospermopsin producers (assay cyrJ) were detected in three lake biofilm samples (Lake Bled, Sava River), while potential saxitoxin producers (assay sxtA) were detected in seven lake biofilm samples (Lake Bled, Koseze Pond) (Table 4, Appendix A). In some samples, only 1/3 or 2/3 technical replicates were positive; all of these samples were close to LOD of the assay. This means that the target genes were present in low quantities and thus not amplified in every subsample (stochastic effect).

### 2.3. Temporal Variability of Microcystin Abundance

Cylindrospermopsins or saxitoxins were not detected with LC-MS/MS in any of the samples. Microcystins were detected in 16 out of 23 plankton samples (out of which one was uncertain as it was too close to the background noise) and 4 out of 22 biofilm samples (out of which 3 were uncertain) with 5 different variants observed (MC-RR, MC-RRdm, MC-HtyRdm, MC-LRdm, MC-LR). The highest concentrations of microcystins were measured in plankton samples from Lake Bled during winter months (up to 660 ng/L in February, BL1.2, Appendix A), which complies with the highest cell concentration of the microcystin-producing species *Planktothrix rubescens* (See Section 2.5. Correlation between Parametres). Moreover, all three microcystin variants found in these samples are typically produced by *Planktothrix rubescens*. Figure 1 shows the microcystin diversity in plankton samples of Lake Bled, where a temporal trend can be observed. In all other samples, microcystins were either not detected or their concentrations were low; up to 6.9 ng/L in plankton samples (SL2, Appendix A) and up to 11.9 ng/g dry weight in biofilm samples (BL2.6, Appendix A) and thus their diversity is not represented in the Figure 1.

### 2.4. Microscopic Analyses

With microscopic analyses, we found 17 potentially toxic taxa in plankton samples (Aphanizomenon sp., Aphanizomenon flos-aquae, Aphanizomenon issatschenkoi, Cylindrospermopsis raciborskii, Dolichospermum crassum, Dolichospermum flos-aquae, Dolichospermum lemmermannii, Dolichospermum planctonicum, Microcystis aeruginosa, Microcystis flos-aquae, Phormidium sp., Phormidium amoenum, Planktothrix agardhii, Planktothrix rubescens, Pseudanabaena sp., Pseudanabaena catenata, Pseudanabaena limnetica) and 4 in biofilm samples (Oscillatoria sp., Phormidium sp., Phormidium autumnale, Pseudanabaena catenata) (Figure 2, Appendix A). The highest diversity of potentially toxic planktic cyanobacteria was found in Lake Pernica (13 taxa), while in Lake Bohinj none was detected. In the majority of the lakes, there was a higher diversity of potentially toxic taxa observed in summer months (June, July, August) than in the rest of the year. Out of the 23 biofilm samples, in the majority of them there was one potentially toxic taxon detected under microscope, while six of them contained none and two of them contained two potentially toxic taxa (Koseze Pond, Lake Bled). The most common potentially toxic genus was Phormidium, which was detected in 17 samples.

### 2.5. Correlation between Parametres

To further evaluate qPCR as a method to detect cyanotoxin production potential, correlations (presented as Pearson correlation coefficient) between qPCR, LC-MS/MS and microscopy results were determined. For plankton samples (N = 25), the numbers of *mcyE* gene copies (sum of values produced by assays mcyE-Ana, mcyE-Mic and mcyE-Pla) were positively correlated with microcystin concentrations measured by LC-MS/MS (r = 0.8375, Figure 3A), while there was no correlation with cell number or biovolume of all potential microcystin-producing taxa. There was also no correlation found between *Microcystis*- or *Planktothrix*-specific *mcyE* gene copies and cell numbers or biovolumes of *Microcystis* or *Planktothrix* cells, respectively. However, when results from Lake Bled were analysed separately, there was a positive correlation between *Planktothrix*-specific *mcyE* gene copies and both cell numbers and biovolumes of *Planktothrix* cells (r = 0.8831 in both cases, the former one is presented on Figure 3B). More detailed graphical representation of results from Lake Bled produced with different methods (Figure 4) shows similar temporal trend observed with qPCR, microscopy and LC-MS/MS. Elevated abundances of *Planktothrix* cells and microcystin concentrations in winter months correspond to a scarlet-coloured blooms of *Planktothrix rubescens* (Figure 2), which were observed on Lake Bled in February 2019 and January 2020.

For plankton samples, correlations could not be determined for assays mcyE-Ana, cyrJ and sxtA as we did not detect target genes in any of the plankton samples. For biofilm samples (N = 23), correlations between target gene copies and relative species abundance were determined and there was no correlation found for any of the assays. Species abundance was evaluated semi-quantitatively (values 1–5), which might have impacted the results.

## 3. Discussion

The study aimed to evaluate the suitability of a qPCR method for early detection of potentially toxic cyanobacteria in surface water bodies in the central European region. Systematic search for publications describing molecular assays (Appendix A) revealed several qPCR assays that were applied for detection of cyanobacteria-specific target genes. We performed a selection of the amplicons, where we took into consideration their specificity, sensitivity and suitability for qPCR reaction conditions. Furthermore, we tested the performance of selected assays (Table 1) in vitro. Even though all assays showed good performance in pure cultures (Table 2) and synthetic DNA fragments, some of them showed unspecific amplification in environmental samples. This is a consequence of heterogenous samples originating from water bodies and presents a high risk in application of SYBR Green chemistry detection in environmental samples, especially when the genome of the target organism is unknown. Nevertheless, we were able to filter the true positive samples from the unspecific samples with reference Tm values. Based on these results, all five selected cyanotoxin-specific assays are suitable for detection of cyanotoxin potential in water bodies. However, as we did not detect cylindrospermopsins or saxitoxins in any of the environmental samples, further research is needed to confirm the suitability of the assays for potential producers of these cyanotoxins. On the other hand, assay 16S-cyano was shown to be inappropriate for detection of cyanobacteria, since it also amplifies plant chloroplast DNA (Appendix A). Thus, we used it as a DNA extraction and qPCR inhibition control.

The qPCR results revealed some new information that was unknown up to date and could not be obtained by microscopy alone. The most novel finding is the potential for cylindrospermopsin- and saxitoxin-production in biofilm in certain water bodies (Table 4), where it has never been reported before. In Slovenia, cylindrospermopsins have been detected once in low amounts in a planktic sample (data not published), while saxitoxins have never been detected. Our study indicates that despite these cyanotoxins not being detected in the moment of sampling (Appendix A), a potential for their production exists, which might be important information for the future monitoring schemes and research studies.

Another important finding is the discovery of cyanotoxic potential in biofilm (Table 4). Even though the first observation of potentially toxic cyanobacteria in biofilm samples was reported in 1997 [24] and cyanotoxins from benthic cyanobacteria are believed to have caused animal death on a few occasions [9,10,11,12,13,24], such studies are still scarce. These findings suggest anatoxin-a and microcytins as the prevalent cyanotoxins in cyanobacterial mats. However, our results indicate such microbial mats might also possess cylindrospermopsin- and saxitoxin-producing potential (found in over a third of the biofilm samples, Table 4), which could not be detected by either microscopy or LC-MS/MS. Although there have been some prior publications about cylindrospermopsin- [25,26,27] and saxitoxin-production [28,29,30] by benthic cyanobacteria, the issue is still poorly investigated. This information, together with some novel findings regarding microcystins in Slovenia (presence of potentially toxic *Microcystis* in Lake Slivnica and Vogrscek throughout the whole year, which has not been reported by regular monitoring before) might be a valuable guideline for future water management.

In addition to applying the qPCR method to environmental samples, our aim was also to compare its performance to traditionally used methods. While many studies have compared qPCR results with cyanotoxins measurements (e.g., [5,6,31]), comparisons with microscopic counts are harder to find, so one of our goals was to evaluate qPCR in comparison with microscopy-based biomonitoring methods. There was no correlation found between gene copy numbers and cell numbers or biovolumes for microcystin-producing cyanobacterial species, neither when analysed separately by genus nor as a whole group. Regarding *Microcystis* genus, there were four samples where *Microcystis*-specific *mcyE* genes were detected, while *Microcystis* cells were not observed under microscope (Appendix A). This could mean that the numbers of *Microcystis* cells in these lakes were below LOD of microscopy, but could be detected by qPCR, which is expected due to much higher sensitivity of qPCR. Alternatively, discrepancy between results could be caused by cross-contamination of field equipment while transferring it between lakes or by low specificity of the assay mcyE-Mic (detecting *mcyE* genes in other genera or other non-target products). The possibility of non-target detection was confirmed also by BLAST analysis, revealing *Pseudanabaena* sp. as one of the assay’s potential targets. However, the majority of the results cannot be explained by this, as *Pseudanabaena* was microscopically observed only in two of these samples. This could be further investigated by DNA sequencing of obtained qPCR products. On the other hand, there were also two samples where *Microcystis* cells were microscopically identified, while *Microcystis*-specific *mcyE* genes were not detected (Appendix A). This might be due to the fact that toxic and non-toxic *Microcystis* cells cannot be distinguished morphologically [32].

For the *Planktothrix* species, elevated cell and microcystin concentrations in Lake Bled in the beginning of the year (Figure 4) correspond to a moderate *Planktothrix rubescens* bloom in 2019. Elevated concentrations at the end of the year contributed to a massive bloom formation that occurred in the end of January 2020 (field observations), which was influenced also by other nutritional factors taking place that month, so it cannot be explained only by our measurement in 2019. Despite the lack of correlation between microscopic and qPCR-based abundance of *Planktothrix* cells in the whole dataset, there was a positive correlation when the analysis was performed only with samples from Lake Bled (Figure 3). Those 11 samples represent a majority of qPCR-positive samples for this assay (mcyE-Pla; Table 4), which indicates that the lack of correlation in other samples is primarily caused by samples with negative qPCR and positive microscopy results. In most of such samples, the dominant *Planktothrix* species was *P. agardhii* (Lake Pernica, Appendix A), which might suggest that the assay does not amplify target genes in the whole genus equally. Alternatively, the discrepancy might be caused by the presence of non-toxic *P. agardhii* strains and the inability of microscopic analyses to differentiate between them, which has been shown in previous studies [33]. Regarding *Dolichospermum* genus, the complete lack of genus-specific *mcyE* genes in all analysed samples (assay mcyE-Ana) despite some microscopic observations (Lake Pernica, Lake Bled; Appendix A) might indicate that the assay does not amplify target genes in all *Dolichospermum* strains, or that the present taxa were in fact not possessing *mcyE* genes.

Furthermore, qPCR was also compared to LC-MS/MS results. A positive correlation was found between *mcyE* gene copy numbers (sum of all analysed genera) and microcystin concentrations (Figure 3), which corroborate numerous prior studies (e.g., [16,31,34]). However, there were also some discrepancies between qPCR and LC-MS/MS results. In some samples, target genes were detected (mostly below LOQ), while cyanotoxins were not (Appendix A). Similar inconsistencies have been observed in other studies as well (e.g., [6,35].) Authors’ potential explanations include low concentration of cyanotoxins (below LOD of analytical method), degradation of cyanotoxins in the samples, lack of gene expression or mutations leading to non-toxicity. It has to be noted that these results are not always expected to match, as analytical methods (such as LC-MS/MS) measure the actual cyanotoxin concentration in a particular moment of sampling, while qPCR detects only the potential for cyanotoxin production. It has been indicated that despite the presence of *mcy* genes, their expression can vary in time significantly [36]. The toxin production depends on physical parameters (e.g., temperature), growth phase [37] and nutrient content [38]. Therefore, quantification of gene copies alone cannot always predict toxin concentration.

Regarding the methodology itself, our study confirmed that qPCR has significantly higher sensitivity (LOD = 1.5–205.9 cells/mL, Table 3) than microscopic cell count (Bürker Türk counting chamber, LOD = 10.000 cells/mL) of Slovenian samples, which are not pre-concentrated. Detection of less than 1 cell/mL (assay sxtA) could be explained by multiple gene copies of the target gene per cell [6] and possible free DNA in the sample. Specificity of all assays was tested and proved appropriate in the original publications. On top of that, our experiments showed that the assays are highly specific in cyanobacterial monocultures (Table 2), while there was some non-specific amplification observed in environmental samples—especially with assays mcyE-Ana and mcyE-Mic. Some of these amplicons are probably primer dimers (Tm < 70 °C), which were observed also in the original study [16]. In order to eliminate false positives, the authors measured fluorescence at a temperature higher than Tm of primer dimers (77 °C). On the other hand, we also observed non-target amplicons with higher Tm (mostly > 80 °C), which indicates amplification of non-target regions. These samples did not show distinctive Tm peaks, but rather multiple smaller peaks, and we confirmed the presence of various non-specific amplicons also by gel electrophoresis. Cq values of such samples were therefore a product of amplification of various templates and could not be considered positive. These false positive signals were excluded from further analyses. The results suggest that the SYBR Green chemistry might not be the most suitable for environmental samples. Specificity and quantification could be improved by using TaqMan chemistry (Roche Molecular Systems Inc., USA) with fluorescent probes or by complementing qPCR results with sequencing of the products.

In some of the assays, amplification efficiency was relatively low (Table 3). While in the original studies amplification efficiency exceeded 90% [6,16,18] for all evaluated assays, in our study that was the case only for assay sxtA. This difference might be caused by sequence variability of uncharacterised cyanobacterial cultures, which is even more significant between different geographical regions of sampling. Possible mismatches in the target regions due to high sequence variability between cyanobacterial strains can lead to low amplification efficiency. Moreover, the effect of the sample impurities and secondary structure of genomic DNA has to be considered as well. Therefore, the LOD and LOQ values, as well as calculated gene copy numbers and cell numbers, might not be fully reliable and should be taken only as a qualitative observation. The reliability of the assay mcyE-Pla might be additionally decreased by a narrow linear dynamic range (Appendix A), which should be addressed in future studies. Moreover, the efficiency of DNA extraction from environmental samples should be evaluated for a more accurate quantification of cells and comparability of results.

For implementation in existing monitoring programs, it is important to quantify cells of the target organisms, not merely gene copies. This might be uncertain in the phylum of cyanobacteria due to unknown number of target gene copies per cell. Genetic cluster *mcy* is thought to appear in only one copy per genome [39,40], while gene *sxtA* appears on average in 3.58 copies per cell in strain *A. circinalis* AWQC131C [6]. Besides, cyanobacteria can contain up to 10 or even more copies of genome per cell [41]; ploidy level differs between species and strains, while it depends also on the growth phase and environmental parameters [42,43]. Therefore, it has to be noted that the calculated cell numbers represent an average for all the genotypes containing target genes from environmental samples, estimated based on assumption that they contain the same number of gene copies as the reference strains. This issue could be avoided if the risk assessment guidelines were adapted for operation with number of gene copies instead of cells.

To enable a thorough cyanotoxin risk assessment in regular monitoring, an assay for detection of anatoxin-a production potential should be designed and optimised. In this study, anatoxin-a was excluded, because literature search did not reveal any appropriate qPCR assays for detection of all anatoxin-producing cyanobacteria. What is more, it would be beneficial to optimise a single assay for detection of *mcyE* genes in all potential microcystin-producers instead of genus-specific assays in order to simplify the test and lower the costs.

## 4. Conclusions

This is the first study in Slovenia aiming to detect cyanotoxic potential with qPCR, as well as one of the few studies employing this method in environmental biofilm samples. We conclude that the method is appropriate for detection of potentially toxic cyanobacteria in water bodies for the purpose of rapid screening and early warning, which could improve risk assessment and protection of human and ecosystem health. Its advantages are early risk detection, short time of analysis and cost effectiveness, while the main downside of the tested assays is suboptimal specificity in environmental samples as a result of SYBR Green chemistry used.

In particular, we aimed to expand the evaluation of the qPCR method also to understudied benthic cyanobacteria in biofilm samples, with the emphasis on comparison with microscopy and LC-MS/MS. The study demonstrated that in both plankton and biofilm samples there might be a potential for cyanotoxin production which is left undetected by traditional methods. This might be especially important in urban water bodies with regular human and animal visitors. In such water bodies, qPCR could provide additional information if implemented in biomonitoring programs, ensuring appropriate precautions to avoid negative effects of acute and chronic exposure to cyanotoxins.

Furthermore, the study indicated that microscopy as the preferred and often the only method of regular biomonitoring is not sufficient for detecting cyanotoxic potential. Similarly, LC-MS/MS did not detect cyanotoxins in all the samples with observed potential for their production. Implementation of the qPCR method with monitoring strategies could serve for assessing potential toxicity of cyanobacterial blooms or microbial mats and forming recommendations for visitors, as well as in evaluating the efficiency of implemented measures for removal or destruction of cyanobacterial cells or cyanotoxins.

## 5. Methods and Materials

### 5.1. Cyanobacterial Cultures and Synthetic DNA Fragments

For specificity testing and cell quantification, reference cyanobacterial cultures were used: microcystin-producing *Anabaena* sp. UHCC 0315 (University of Helsinki, Finland) [16], *Microcystis aeruginosa* PCC 7806 (Pasteur Institute, France) [16] and *Planktothrix agardhii* NIVA-CYA 126 (Norwegian Institute for Water Research, Norway) [17], cylindrospermopsin-producing *Aphanizomenon ovalisporum* ILC-164 (Israel Oceanographic and Limnological Research, Israel) [44] and saxitoxin-producing *Aphanizomenon gracile* NIVA-CYA 851 (Norwegian Institute for Water Research, Norway) [45]. The cultures were grown in standard medium BG11 (Gibco, ThermoFisher Scientific, Waltham, MA, USA) under natural light conditions at room temperature. For DNA extraction (see Section 5.3. DNA Extraction and Quality Control), the cultures were filtered through Sterivex columns (Milipore Sterivex-GP Pressure Filter Unit, Merck KGaA, Germany) with 0.22 µm pore size; the volume filtered was between 5 and 19 mL. We estimated cell concentration using Bürker Türk counting chamber and light microscopy.

For positive controls of qPCR reactions and gene copy quantification, synthetic DNA fragments (gBlocks, Integrated DNA Technologies, Coralville, IA, USA) specific to the selected target regions were used. Their nucleotide sequences were determined based on reference sequences from NCBI GenBank [46]; *Anabaena* sp. 90 (AJ536156.1) for mcyE-Ana, *Microcystis aeruginosa* PCC 7806 (AF183408.1) for mcyE-Mic, *Planktothrix agardhii* 213 (EU151891.1) for mcyE-Pla, *Aphanizomenon* sp. 10E6 (GQ385961.1) for cyrJ, *Anabaena circinalis* AWQC131C (DQ787201.1) for sxtA and *Anabaena circinalis* AWQC131C (AF247589.1) for 16S-cyano (Appendix A).

### 5.2. Environmental Sampling

In total, 25 plankton and 23 biofilm samples were collected from 7 lakes or reservoirs and 8 rivers or streams in Slovenia in 2019 (Appendix A); one of them (plankton, BL1.4) was included only in cyanotoxin analysis. Plankton samples were collected in lakes with integrating water sampler (Hydro-Bios, IWS III, Germany) 11 times in the pilot area, Lake Bled, and 3–4 times elsewhere. Biofilm samples were collected once in rivers and selected lakes, by brushing biofilm off stones or—if stones were not available—macrophytes, wooden substrate or bricks. Sampling was performed according to national guidelines for monitoring of ecological state of water bodies intercalibrated in the frame of Water Framework Directive implementation. All field equipment used for community DNA samples was treated beforehand with 10% H_2_O_2_ solution and rinsed with distilled water. The sampling procedure was controlled in the field by using blank controls with Milli-Q water (Merck KGaA).

Each sample was collected in three aliquotes for community DNA extraction, microscopic cell count and cyanotoxin analysis. For DNA extraction from plankton samples, 60–1000 mL of water was filtered through Sterivex columns (Milipore Sterivex-GP Pressure Filter Unit, Merck KGaA) with pore size 0.22 µm. The columns were stored on –20 °C for up to 161 days. For biofilm, 10 mL of biofilm-Mili-Q water mixture was mixed with 40 mL of absolute ethanol and stored on 4 °C for up to 147 days. For cyanotoxin analysis, plankton (200–2000 mL) or biofilm (5 or 10 mL) samples were filtered through GF/C filters (Whatman, GE Healthcare, Chicago, IL, USA) with pore size 1.2 µm and dry weight was determined (for biofilm). Filters were stored on –20 °C for up to 14 months before analysed with LC-MS/MS. Cyanobacterial cell count was performed under light microscope with counting chambers (e.g., Hydro-Bios, Germany). For plankton samples, cell numbers were counted, and volumes were calculated for each species (i.e., biovolumes). For filamentous cyanobacterial species, where cell numbers cannot be directly counted, first their biovolumes were calculated based on the measurements of filament length and width, and then this was converted to cell numbers based on species-specific cell biovolumes known from literature or from our previous measurements. For biofilm samples, species abundance was evaluated semi-quantitatively by assigning each species a value of 1–5 based on their abundance (0—not detected, 1—very rare, 2—rare, 3present, 4—frequent, 5—dominant).

### 5.3. DNA Extraction and Quality Control

DNA extraction was performed using commercially available kits following manufacturers’ instructions; for cyanobacterial cultures and plankton samples DNeasy PowerWater Sterivex kit (Qiagen, Germany) and for biofilm samples NucleoSpin Soil kits (Macherey-Nagel, Germany) were used. DNA was stored at –20 °C for further analyses. In each extraction, a blank control using sterile water (B. Braun, Germany) was included. DNA concentration and purity of the samples and blank controls were evaluated using spectrophotometer NanoDrop (Thermo Scientific, ThermoFisher Scientific) with 1.5–2 µL of DNA sample and elution buffers from DNA extraction kits as a background.

### 5.4. qPCR Setup, Assay Validation and Quantification

After a literature review of previously designed qPCR assays for the detection of potentially toxic cyanobacteria, nine published assays were selected and evaluated based on their performance described in the published papers and in silico and in vitro characterisation in this study. In addition to the specificity evaluation done in the original papers (in silico and in vitro for all primers), in silico specificity check was performed with NCBI Primer-BLAST [47] using nr/nt database. Furthermore, specificity was tested in vitro on five different cyanobacterial cultures (see below) for all assays, and for assay 16S-cyano additionally with selected heterotrophic bacterial strains that could be present in freshwater habitats (*Salmonella enterica*, *Escherichia coli*, *Pseudomonas fluorescens*, *Brevundimonas* sp., *Arcobacter butzleri*) and with selected plant samples to check for amplification of plant chloroplasts (*Solanum lycopersicum*, *Vitis vinifera*, *Alnus glutinosa*, *Clematis* sp.). The final set of assays included five qPCR assays to target cyanotoxin-producing cyanobacteria and an additional assay to target 16S rRNA gene, which served as a DNA extraction and qPCR inhibition control (Table 1). Amplification was performed on qPCR cycler Applied Biosystems 7900HT (ThermoFisher Scientific). First, different primer concentrations (0.3 µM and 0.9 µM) and reaction volumes (10 µL and 20 µL) were tested and optimised for each assay. Final reaction volume was 10 µL, consisting of 5 µL of SYBR Green PCR Master Mix (Applied Biosystems, ThermoFisher Scientific), 0.9 µM (assays mcyE-Ana, mcyE-Pla, cyrJ, sxtA and 16S-cyano) or 0.3 µM (assay mcyE-Mic) of each primer and 2 µL of DNA template in 10^−1^ dilution. Potential for qPCR inhibition was evaluated on up to five selected samples for assays mcyE-Pla, cyrJ and sxtA, by analysing two subsequent dilutions (10^−1^ and 10^−2^) for each sample. Reactions were performed in 384-well clear PCR plates (Thermo Scientific, ThermoFisher Scientific), covered by MicroAmp™ optical adhesive film (Applied Biosystems, ThermoFisher Scientific). Temperature profile was as follows: 2 min on 50 °C, 10 min on 95 °C, followed by 45 cycles of 15 s on 95 °C and 1 min on 60 °C. Dissociation stage with initial denaturation for 15 s on 95 °C, followed by 15 s on 60 °C and a gradual increase up to 95 °C, was added at the end. Every reaction was performed in three technical replicates. Positive (specific synthetic DNA fragments) and negative controls (nuclease-free water, Sigma-Aldrich, St. Louis, MS, USA) were included in every experiment. qPCR amplification conditions were the same for synthetic DNA and DNA isolated from cultures or environmental samples.

qPCR results were analysed with software SDS (version 2.4.1, Applied Biosystems, ThermoFisher Scientific). Threshold values were set manually for each assay within linear part of exponential curve, allowing for the comparison of Cq values between runs. For each sample, the amplification curve (giving Cq value) and dissociation curve of the amplified product (giving Tm value) was checked. To exclude false positives, results were considered positive if the following two criteria were met: Cq was within detection range and there was a distinctive peak with appropriate Tm. Cut-off values for Cq (values at the highest dilution within detection range) and reference values for Tm (average Tm values of all DNA dilutions within quantification range) were determined based on results from cyanobacterial cultures. Tm values served to control the specificity of amplification, therefore only values in the expected range were considered positive.

Samples that resulted in multiple peaks with one of them showing appropriate Tm were further analysed with agarose gel electrophoresis. qPCR products were run on 2% agarose gel with 1× Tris Acetate-EDTA (TAE) buffer stained with ethidium bromide at 100 V for 90 min. Samples contained 4 μL of each product and 1 μL of 6× Mass Ruler DNA Loading Dye (Thermo Scientific, ThermoFisher Scientific), and GeneRuler 100 bp DNA Ladder (Thermo Scientific, ThermoFisher Scientific) was used to determine the product length. Visualisation of amplicons was performed under UV light using UVP ChemStudio PLUS Imaging System (Analytik Jena, Germany).

Assay sensitivity (LOD, LOQ, linear dynamic range) of the assays was evaluated from the cyanobacterial culture calibration curves. LOD was defined as a number of cells where at least 2/3 technical replicates produced a positive result. LOQ was defined as number of cells in sample that gave Cq value at the lower end of the linear curve and the CV of technical replicates did not exceed 2%. Amplification efficiency (e) was calculated by the equation e = 10^−1/S^—1, where S represents the slope of the linear dynamic range of the calibration curve. Potential for qPCR inhibition was evaluated by comparing differences between average Cq values of two subsequent dilutions for the same sample.

For results with 3/3 positive technical replicates, average Cq value (which was the basis for quantification of cells and gene copies) and absolute difference between Cq values of technical replicates (which served for assessing intra-assay variability) were calculated. Quantification was performed using calibration curves approach. The calibration curves were generated from eight subsequent dilutions of cyanobacterial cultures DNA for quantification of target cells, or of synthetic DNA fragments for quantification of target gene copies. DNA concentration of stock solution of DNA fragments was 10 ng/µL (according to the manufacturer), from which gene copy numbers were calculated based on the following equation:$$gene\;copy\;number=\;\frac{DNA\;amount\left[ng\right]\times 6.022\times 10^{23}}{DNA\;fragment\;lenght\left[bp\right]\times 650\times 10^{9}}$$

Results with calculated values below LOQ were given a value of LOQ/2. Results with 1/3 or 2/3 positive technical replicates were given a value of LOQ/10. These values were used for all following analyses. To ensure comparability of results despite variable volumes of plankton samples filtered prior to DNA extraction and variable densities of biofilm samples, gene copies per microliter of DNA were converted into values per millilitre of water (plankton) or gram of dry weight (biofilm).

Intra-assay variability was tested by using three technical replicates within each experiment. Repeatability (inter-assay variability) of qPCR reactions was tested by repeating the experiment two times with the same template DNA and under the same conditions on up to five selected samples for assays mcyE-Pla, cyrJ and sxtA, and comparing average Cq values of the two technical repetitions. Robustness of the assays was evaluated by testing them on diverse set of samples; cyanobacterial monocultures, synthetic DNA fragments, frozen or lyophilised cyanobacterial bloom samples and environmental samples (plankton, biofilm).

### 5.5. Cyanotoxin Analysis with LC-MS/MS

Intracellular cyanotoxins were extracted from filters applying the protocol described by Cerasino and Salmaso [48] and quantified with LC-MS/MS. The extraction was carried out by using a mixture of acetonitrile in water (60/40 v/v), containing 0.1% formic acid. Extracted toxins were injected into a LC-MS/MS system, composed of a Waters Acquity UPLC system (Waters, Milford, MA, USA) coupled to a SCIEX 4000 QTRAP mass spectrometer (AB Sciex Pte. Ltd., Singapore). The mass detector was operated in scheduled MRM (Multiple Reaction Monitor) mode, using positive electrospray ionisation (ESI+). Quantification of microcystins was performed following the protocol from Cerasino and Salmaso [48], which has the capability of determining the 11 most common microcystin variants, namely RR, [D-Asp3]-RR (RRdm), [D-Asp3]-HtyrR (HtyRdm), YR, LR, [D-Asp3]-LR (LRdm), WR, LA, LY, LW, LF. Analysis of cylindrospermopsins and saxitoxins was performed following the protocol from Ballot et al. [49], targeting CYN, STX, dcSTX, NeoSTX, GTX1, GTX4, GTX5, C1 and C2.

### 5.6. Data Analysis

Calibration curves and their Pearson correlation coefficients were prepared in Microsoft Excel (2007). Correlations and linear regression curves between gene copy numbers and microcystin concentrations, cell numbers and biovolumes were determined with Pearson correlation coefficient, using Prism 6 (GraphPad Inc., San Diego, CA, USA) with 95% confidence interval. For biofilm samples, relative abundance was calculated by summing up the values of individual target species and normalising the summed value to 100%. In order to include negative values (below LOD) in correlation analysis, they were replaced with a minimum detected value for each assay divided by 100 (for microscopy and LC-MS/MS results) or by 10 (for qPCR results).

## Figures and Tables

**Figure 1 toxins-13-00133-f001:**
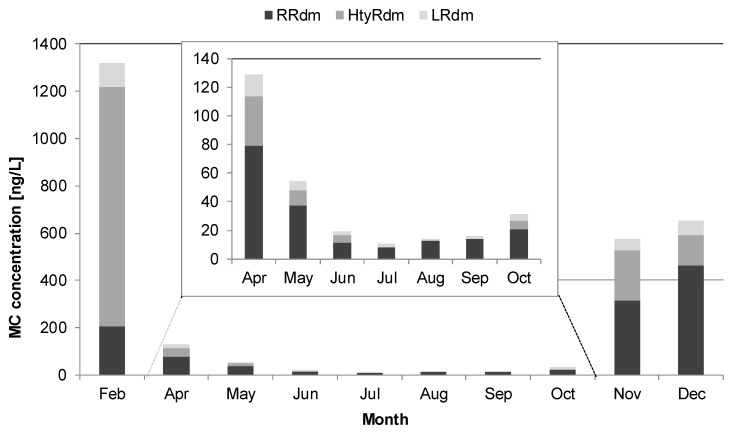
Microcystin temporal variability and diversity in plankton samples from Lake Bled. The three congeners (RRdm, HtyRdm and LRdm) are all demethylated variants, typical of *Planktothrix rubescens*. The temporal changes of proportions can be explained by the succeeding of different chemotypes of the same species. Samples Feb–Dec correspond to samples BL1.2–BL1.12 (Appendix A).

**Figure 2 toxins-13-00133-f002:**
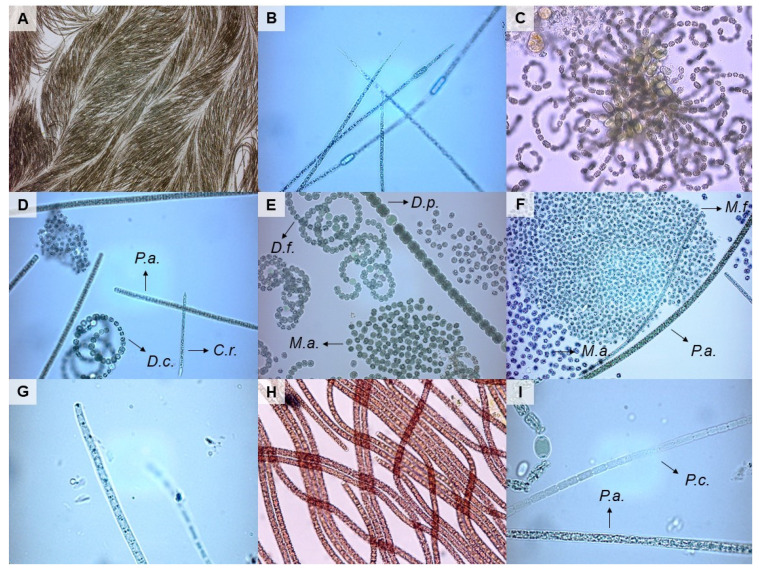
Potentially toxic cyanobacterial taxa found in environmental samples. (**A**)—Aphanizomenon flos-aquae, (**B**)—Aphanizomenon issatschenkoi, (**C**)—Dolichospermum lemmermanii, (**D**)—C.r. Cylindrospermpsis raciborskii, D.c. Dolichospermum crassum, P.a. Planktothrix agardhii, (**E**)—D.f. Dolichorpermum flos-aquae, D.p. Dolichospermum planctonicum, M.a. Microcystis aeruginosa, (**F**)—M.a. Microcystis aeruginosa, M.f. Microcystis flos-aqaue, P.a. Planktothrix agardhii, (**G**)—Phormidium amoenum, (**H**)—Planktothrix rubescens, (**I**)—P.a. Planktothrix agardhii, P.c. Pseudoanabaena catenata. The photos were taken under light microscope with 160× (**A**), 400× (**C**,**H**), 640× (**B**,**D**,**E**,**F**) or 1600× magnification (**G**,**I**).

**Figure 3 toxins-13-00133-f003:**
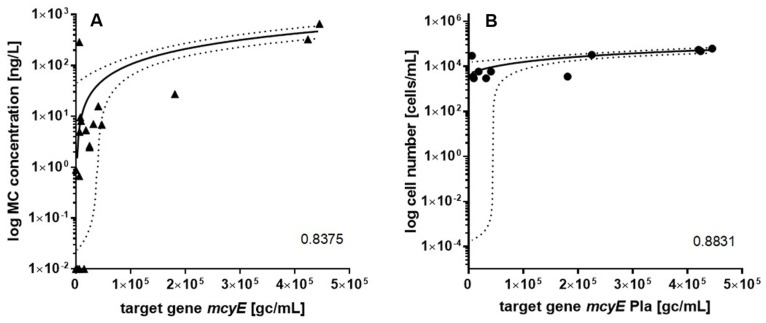
Scatterplots showing the correlations between different parameters for plankton samples. Solid lines represent linear regression curves, dotted lines represent 95% confidence band, and Pearson correlation coefficient (r) is given in the bottom right-hand corners. (**A**) correlation between *mcyE* copies concentration (the sum of concentrations obtained by assays mcyE-Ana, mcyE-Mic and mcyE-Pla) and MC concentration. (**B**) correlation between *Planktothrix*-specific *mcyE* gene copies concentration and cell concentration of all *Planktothrix* species (only samples from Lake Bled, N = 11).

**Figure 4 toxins-13-00133-f004:**
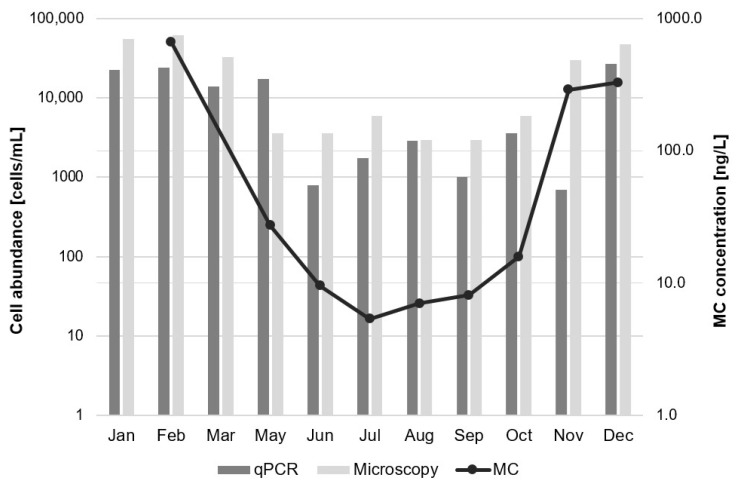
*Planktothrix* cell abundance in Lake Bled in 2019 determined by qPCR (assay mcyE-Pla) and microscopic analyses. Due to low amplification efficiency and possible variability of gene copy numbers per cell, the calculated cell abundance values are not reliable and should be used as qualitative observation. Total MC concentration measured by LC-MS/MS is included for comparison. In microscopy, 10% variation is expected, which is the average standard deviation between technical replicates with Bürker Türk counting chamber in our laboratory.

**Table 1 toxins-13-00133-t001:** Primers used for qPCR amplification of selected target regions. bp—base pair.

Target Cyanotoxins/Organisms	Assay	Target Gene	Primer Label	Nucleotide Sequence (5′ → 3′)	Fragment Length [bp]	Reference
Microcystins (genus *Dolichospermum*)	mcyE-Ana	*mcyE*	mcyE-F2	GAA ATT TGT GTA GAA GGT GC	247	[15]
			AnamcyE-12R	CAA TCT CGG TAT AGC GGC		[16]
Microcystins (genus *Microcystis*)	mcyE-Mic	*mcyE*	mcyE-F2	GAA ATT TGT GTA GAA GGT GC	247	[15]
			MicmcyE-R8	CAA TGG GAG CAT AAC GAG		[16]
Microcystins (genus *Planktothrix*)	mcyE-Pla	*mcyE*	mcyE-F2	GAA ATT TGT GTA GAA GGT GC	249	[15]
			PlamcyE-R3	CTC AAT CTG AGG ATA ACG AT		[17]
Cylindrospermopsins	cyrJ	*cyrJ*	cyrJ207-F	CCC CTA CAA CCT GAC AAA GCT T	77	[18]
			cyrJ207-R	CCC GCC TGT CAT AGA TGC A		
Saxitoxins	sxtA	*sxtA*	sxtA-F	GAT GAC GGA GTA TTT GAA GC	125	[6]
			sxtA-R	CTG CAT CTT CTG GAC GGT AA		
(Cyano-)bacteria and plant chloroplasts	16S-cyano	16S rRNA	cyano-real16S-F	AGC CAC ACT GGG ACT GAG ACA	73	[6]
			cyano-real16S-R	TCG CCC ATT GCG GAA A		

**Table 2 toxins-13-00133-t002:** Specificity of selected qPCR assays, with positive results shaded in grey. Average quantification cycle (Cq) values between three technical replicates of DNA in 10^−2^ dilution and reference melting temperatures (Tm, in °C), calculated as an average of all DNA dilutions within quantification range are shown. Raw data is available in Appendix A. MC—microcystins, CYN—cylindrospermopsins, SXT—saxitoxins, N—no specific amplification.

Cyanobacterial Cultures		qPCR Assays											
Strain	Toxicity	16S-cyano		mcyE-Ana		mcyE-Mic		mcyE-Pla		cyrJ		sxtA	
		Cq	Tm	Cq	Tm	Cq	Tm	Cq	Tm	Cq	Tm	Cq	Tm
*Anabaena* sp. UHCC 0315	MC-producer	23.13	81.1	25.86	75.7	N	N	N	N	N	N	N	N
*Microcystis aeruginosa* PCC 7806	MC-producer	20.14	81.0	N	N	19.82	78.0	N	N	N	N	N	N
*Planktothrix* sp. NIVA-CYA126/8	MC-producer	23.55	81.1	N	N	N	N	24.37	77.6	N	N	N	N
*Aphanizomenon ovalisporum* ILC-164	CYN-producer	23.87	81.2	N	N	N	N	N	N	24.79	80.0	N	N
*Aphanizomenon gracile* NIVA-CYA 851	SXT-producer	18.79	81.3	N	N	N	N	N	N	N	N	18.72	79.6
	Reference Tm	81.1		75.9		78.2		77.6		79.9		79.5	

**Table 3 toxins-13-00133-t003:** Limit of detection (LOD), limit of quantification (LOQ), amplification efficiency and correlation coefficient for selected qPCR assays based on calibration curves of reference cyanobacterial strains DNA. LOQ and LOD are expressed in cells/µL DNA and in cells/mL sample, which depends on the individual sample volume. Individual calibration curves are shown in Appendix A.

Assay	LOD (Cells/µL DNA, Cells/mL Sample)	LOQ (Cells/µL DNA, Cells/mL Sample)	Amplification Efficiency	Correlation Coefficient (R^2^) of Linear Range of Curve
mcyE-Ana	3.3, 30.1	33, 301	0.63	1.0000
mcyE-Mic	16.5, 205.9	165, 2059	0.75	0.9973
mcyE-Pla	2.4, 12.5	238, 1252	0.68	0.9969
cyrJ	5.7, 114.6	6, 115	0.73	0.9991
sxtA	0.3, 1.5	3, 15	0.98	0.9988
16S-cyano	1.6, 20.6	165, 2059	0.79	0.9968

**Table 4 toxins-13-00133-t004:** Abundance of target gene copies in environmental samples. Plankton [gc/mL]: * 1–10^2^, ** 10^2^–10^4^, *** 10^4^–10^6^; biofilm [gc/g dry weight]: * 10^3^–10^5^, ** 10^5^–10^7^. Due to low amplification efficiency, the gc values are not reliable and should be used as qualitative observation. Sample description and quantification data is available in Appendix A. Sample BL1.4 has only been analysed for cyanotoxin content and not by qPCR and is thus not represented in this table. gc—gene copies, empty—below LOD.

		qPCR Assays				
	Sample	mcyE-Ana	mcyE-Mic	mcyE-Pla	cyrJ	sxtA
PLANKTON [gc/mL]	BL1.1			***		
	BL1.2			***		
	BL1.3			***		
	BL1.5			***		
	BL1.6			**		
	BL1.7			***		
	BL1.8			***		
	BL1.9			**		
	BL1.10			***		
	BL1.11			**		
	BL1.12			***		
	BO1					
	BO2					
	BO3			*		
	PE1		***			
	PE2		***			
	PE3		***			
	PE4		**			
	SL1		**			
	SL2		***			
	SL3		**			
	VO1		**			
	VO2		**			
	VO3		**			
BIOFILM [gc/g ry weight]	BL2.1					**
	BL2.2				*	*
	BL2.3					**
	BL2.4					*
	BL2.5					
	BL2.6				*	
	BL2.7			*		*
	BL2.8					
	BL2.9					
	BL2.10					**
	SO1					
	SO2					
	SO3					
	LU					
	BI					
	PS					
	SA1				*	
	SA2					
	RI					
	LJ					
	KO					**
	TI					
	GL					

## Data Availability

Majority of the data produced in this study is available in Appendix A. Additional data is available upon request to the corresponding author.

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
