# Peer review of "Potentially Toxic Planktic and Benthic Cyanobacteria in Slovenian Freshwater Bodies: Detection by Quantitative PCR"

_toxins, 2021, doi:10.3390/toxins13020133_

Round 1

Reviewer 1 Report

Current MS is devoted to qPCR detection of potentially toxic cyanobacteria in environmental samples in different regions of Slovenia. Using qPCR as tools for screening of cyanotoxin-producing microorganism was mentioned in since 2003. Although the research is well written but there is no new approach or methodology. The use of qPCR for estimating the potential cyanotoxicity of blooms has already been proven in a number of publications: http://dx.doi.org/10.3390/toxins8060172 and http://dx.doi.org/10.1016/j.watres.2017.04.025. I found only one new data - the detection of cyanobacterial toxic potential with qPCR in Slovenia lakes. Current MS can be published in the Toxins journal as short communication with an emphasis on cyantoxins in the Slovenia lakes.

Major concerns

“2.4. Microscopic analyses” section does not contain sufficient information about features of microscopic organization of plankton samples. This section contains only systematic position of the samples. But systematic position authors revealed by PCR. I recommend to remove this section form MS. If Authors prefer to stay this microscopic data in MS – They have to extend “2.4. Microscopic analyses” section, including images for each sample (see my recommendations below), and add table with morphological measurements of the samples (body length, diameter etc.)

 Minor concerns

Line 34-39: “Gaget et al., 2017a” has been misinterpreted by Authors. Please remove or rephrase this sentence.

Line 99-170 contains methodological information – this section have to move to “Material and methods” section

Line 256 – Authors set only one photo on Planktothrix rubescens. I highly recommend set images for other 16 toxic taxa of plankton samples.

Line 313-314: In line 231-232 Authors mentioned that Cylindrospermopsins and saxitoxins were not detected with LC-MS/MS in any samples. For me, Authors cannot discuss about correlation between these toxins (Cylindrospermopsins and saxitoxins) and existing of their genes. In line 313-314 authors should discuss only about detection of Microcystins.

Line 398-400 This sentence is very undoubted. (1) How authors calculated “LOD = 10.000 cells/mL”? (2) For microscopical research the water samples are always concentrated by millipore filters and only after this the samples are researched by microscopic technique. Therefore, it is not appropriate to talk here about LOD of unconcentrated water samples. I recommend to remove or rephrase this sentence.

Line 450-451: Set reference for this sentence (“Firstly, there have been no reports on its presence in Slovenian water bodies so far”)

Reviewer 2 Report

The presented interesting and comprehensive study was devoted to analysis of 25 plankton and 23 biofilms collected from 15 water bodies in Slovenia during one year for cyanotoxin presence and cyanobacteria producers.   The authors applied and compared three experimental methods: (1) qPCR targeting mcyE, cyrJ and sxtA genes involved in cyanotoxin production, (2) LC-MS/MS quantifying microcystin, cylindrospermopsin and saxitoxin concentrations, and (3) light microscopy.

As it is written in the manuscript, the “aim was to evaluate the suitability of those assays for detection of cyanotoxin threat in surface water bodies, optimise them for environmental samples and compare the results with traditional identification methods (microscopy, LCMS/MS).”(Lines 80-83)

 “First, we have evaluated the selected qPCR assays in terms of their specificity, sensitivity and robustness”. Lines 96-97

The main comments are:

“Conclusions” section has to contain the main conclusions on the experiments performed in accordance with the purpose of the study. The authors can add additional information in this section by answering to questions:

What are the strengths and weaknesses of the qPCR method for performed studies?

What was the optimization of the methods carried out by the authors?

What is the scientific novelty of the work?

Please, indicate more clearly the main conclusions from the conducted experiments also in the Abstract section.

The minor comments are:

(1) Figure 1

October has to be instead of Oktober

(2) Section 2.4 The strain names have to be written in Italic.

(3) It is customary to write numbers up to ten in words, not numbers.

 Line 262 “1” has to be replaced by “one”.

Lines 263-263 instead of “2” has to be “two”.

Reviewer 3 Report

This research on the detection of cyanobacteria by quantitative PCR was well designed and well reported in the current manuscript. The full set of data was presented in detail in the supplementary files.

I have only minor issues below:

Line 159: raw data was presented in Fig. S4, not S3

Line 251-258- italicise species names.

Line 601-603- « LOD was defined as a number of cells where at least 2/3 technical replicates produced a positive result. » Ok! However, I do not see here the definition of LOQ.

Round 2

Reviewer 1 Report

All my recomendations/propositions were implemented by authors. Now MS can be publeihs in Toxins journal.